# High-power electrically pumped terahertz topological laser based on a surface metallic Dirac-vortex cavity

Junhong Liu [1,2,4], Yunfei Xu [1,2,4], Rusong Li[3], Yongqiang Sun[1,2], Kaiyao Xin[1,2], Jinchuan Zhang [1,2] ✉, Quanyong Lu [3] ✉, Ning Zhuo[1], Junqi Liu[1,2], Lijun Wang[1,2], Fengmin Cheng[1], Shuman Liu[1,2], Fengqi Liu [1,2] ✉ & Shenqiang Zhai [1] ✉

Topological lasers (TLs) have attracted widespread attention due to their mode robustness against perturbations or defects. Among them, electrically pumped TLs have gained extensive research interest due to their advantages of compact size and easy integration. Nevertheless, limited studies on electrically pumped TLs have been reported in the terahertz (THz) and telecom wavelength ranges with relatively low output powers, causing a wide gap between practical applications. Here, we introduce a surface metallic Dirac-vortex cavity (SMDC) design to solve the difficulty of increasing power for electrically pumped TLs in the THz spectral range. Due to the strong coupling between the SMDC and the active region, robust 2D topological defect lasing modes are obtained. More importantly, enough gain and large radiative efficiency provided by the SMDC bring in the increase of the output power to a maximum peak power of 150 mW which demonstrates the practical application potential of electrically pumped TLs.

In recent years, topological lasers (TLs) have attracted widespread research interest due to their topologically-protected mode robustness from perturbations or defects, which dramatically improves the lasing stability; thus, TLs are becoming promising light source candidates for future photonic integrated chips. To date, various TLs have been demonstrated based on different topological phenomena, such as the quantum Hall effect[1–3], quantum spin Hall effect[4,5], quantum valley Hall effect[6–8], and higher-order topology[9,10]. In addition, lasing in the topological edge or zero mode has been achieved with a 1D Su–Schrieffer–Heeger (SSH) state[11,12], topological corner state[13,14], and 2D topological defect state[15–17]. Specifically, based on the topological zero mode in the middle of the photonic band gap, Lu et al.[18,19] demonstrated a topological cavity surface emitting laser (TCSEL) with excellent performance; this TCSEL exhibits considerable space for performance improvement of TLs. However, all these reported devices are based on optical pumping, where bulky external pumping setups

are needed. Thus, compact electrically pumped TLs that can convert electrical energy directly to light need to be developed.

Limited studies on electrically pumped TLs[20–22] have been reported in the terahertz (THz) and telecom wavelength ranges; moreover, their output powers remain quite low, producing a large gap between practical applications. For example, an estimated maximum peak power of 9.04 mW was demonstrated by ref. 21 in an electrically pumped THz TL based on a photonic analog of a Majorana zero mode (MZM). To attain an electrically pumped TL, efficient carrier injection and large-mode confinement are needed. Therefore, the active regions of all these reported devices were graphically etched to introduce a high refractive index contrast for strong mode confinement. This results in a substantial gain decrease caused by the weakened active region and strong optical scattering due to the uneven surface of the etched sidewalls, causing difficulty for power improvement. Moreover, the deep graphical etching process also complicates the fabrication

[1]Laboratory of Solid-State Optoelectronics Information Technology, Institute of Semiconductors, Chinese Academy of Sciences, Beijing, China. [2]Center of Materials Science and Optoelectronics Engineering, University of Chinese Academy of Sciences, Beijing, China. [3]Division of Quantum Materials and Devices, Beijing Academy of Quantum Information Sciences, Beijing, China. [4]These authors contributed equally: Junhong Liu, Yunfei Xu. ✉e-mail: zhangjinchuan@semi.ac.cn; luqy@baqis.ac.cn; fqliu@semi.ac.cn; zsqlzsmbj@semi.ac.cn

process and increases the device cost, hindering the advancement of practical applications.

Here, we show high-power electrically pumped TLs based on a surface metallic Dirac-vortex cavity (SMDC) design in the THz spectral range. Due to the Dirac-vortex cavity designed and fabricated in the surface metal layer, the devices operate in a robust 2D topological defect lasing mode; this mode is the Jackiw–Rossi zero mode based on the zero-mode solutions of Dirac equations[23] with mass vortices. Owing to the sufficient coupling strength of the double-metal waveguide structure, a band gap width of 3% for the SMDC is numerically simulated, which enables robust topological mode lasing. Furthermore, owing to the surface Dirac-vortex topological cavity design, the active region is nondestructive, ensuring a large enough gain for high output power. THz SMDC TLs demonstrate robust single-mode THz surface emission with a maximum peak power of 150 mW. This value produces an order of magnitude improvement compared with that of previously reported electrically pumped THz TLs considering the same size and is close to the best results of other general types of single-chip surface-emitting THz lasers[24,25] with single-mode operation. In principle, this approach with a topological cavity designed in the surface waveguide layers without interrupting the active region is also applicable to semiconductor lasers in other wavelength ranges, such as near-infrared and mid-infrared wavelengths, as long as sufficient refractive index contrast is achieved. Moreover, adjustable far-field patterns are obtained by introducing a nonuniform phase distribution while keeping the spectral and vector-polarization characteristics unchanged; thus, our study provides an effective and convenient approach to tune the far-field pattern through topological photonics due to the simplified device process.

## Results

### Device structure of the SMDC TLs

A schematic of the proposed THz SMDC TL is depicted in Fig. 1a. Different from the traditional topological cavity design, where graphical etching of the active regions is usually needed, the Dirac-vortex cavity is designed and fabricated in the surface metal layer based on a double-metal waveguide structure with a THz quantum cascade laser (QCL) wafer. The pumped current is provided by three square mesas covered with metal as the positive electrode and the back metal fabricated by electron beam evaporation on the n-type GaAs substrate as the negative electrode. Optical microscopy images of the whole laser chip and scanning electron microscopy (SEM) images of the cavity center and boundary are shown in Fig. 1b. The Dirac-vortex cavity is based on a honeycomb lattice with a hexagonal supercell composed of six neighboring sites. The highly doped layer of the QCL epitaxy structure under the hexagonal mesa is removed, while that in the mesa periphery area is retained as the absorption boundary. This absorption boundary design is introduced to increase the loss of the whispering gallery modes[26] to improve the robustness of the topological mode under electrical pumping conditions. The SEM image of the cavity center area demonstrates some air holes are connected in pairs. This is because the metal between these air holes were stripped off in the metal stripping process due to the small distance between the holes. These fabrication imperfections could have a negligible influence to the device performance for two reasons. Firstly, the connecting areas are rather small and won't affect the general symmetry of the unit cell. Secondly, these connections are only observed in certain area of the cavity with larger values of $m$ for higher coupling strength. Figure 1c, d shows the schematic illustration and SEM image of the device cross-section, respectively, demonstrating the nondestructive active region.

### Theoretical design for the SMDC TLs

First, a honeycomb lattice with a hexagonal supercell composed of six neighboring sites is chosen as the unit cell to construct the topological cavity. Since the original Dirac cones at the K and K' points in the first Brillouin zone of the honeycomb lattice are folded to doubly degenerate Dirac cones at the Γ point, the modes at the Γ point have no in-plane wave vectors, which are conducive to the extraction of surface emission. For the Dirac-vortex cavity design, the key issue is to create a sufficiently wide photonic band gap for the surface topological cavity to ensure 2D topological defect mode lasing. By shifting three spaced sites along a specific displacement vector **m** in the hexagonal

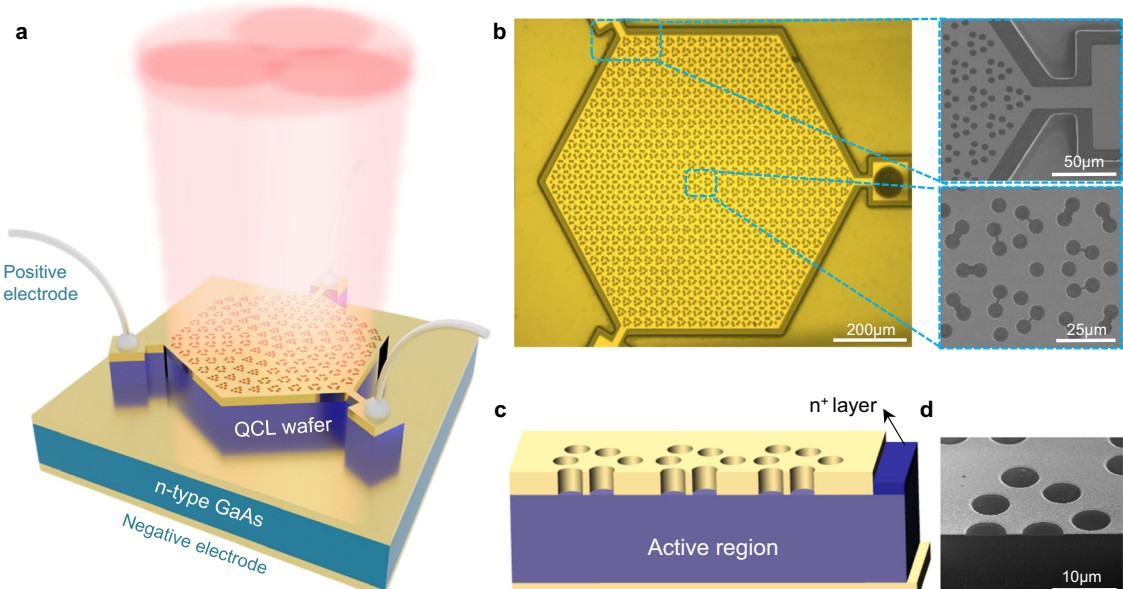

**Fig. 1 | Electrically pumped THz SMDC TL. a** Schematic diagram of the electrically pumped THz SMDC TL. A topological cavity is constructed in the surface metal layer based on a double-metal waveguide structure. The device maintains a hexagonal shape, and three spaced electrodes are introduced for current injection. **b** Optical microscope image of the whole laser chip and scanning electron microscope (SEM) images of the cavity center and absorption boundary. The topological cavity consists of a honeycomb lattice with a hexagonal supercell composed of six neighboring sites. **c** Schematic illustration of the device cross-section. The top n⁺ layer under the hexagonal mesa is removed, while that in the mesa periphery area with a width of 15 μm is retained as the absorption boundary. **d** SEM image of the device cross section obtained by a focused ion beam (FIB).

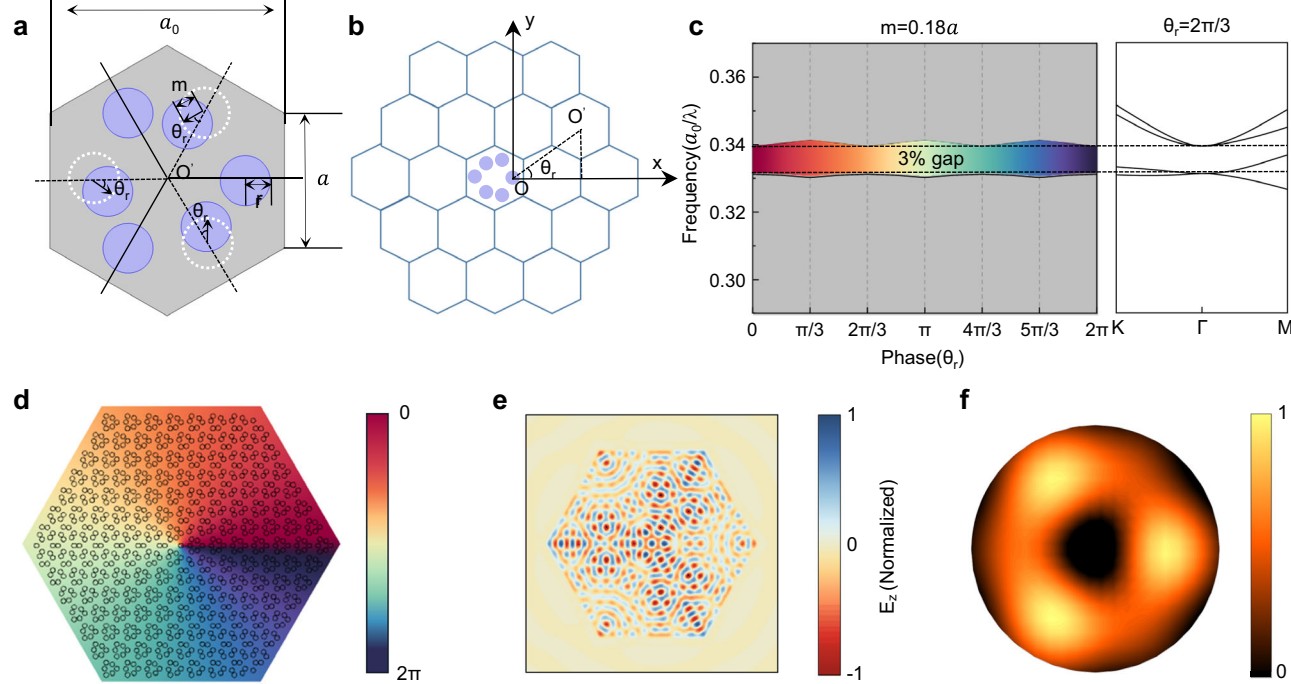

**Fig. 2 | Design of the terahertz topological laser based on the SMDC.**
**a** Honeycomb primitive cell of the Kekulé modulation, with a circular air hole radius $r = a_0/9$. **b** Schematic diagram of the hexagonal topological cavity construction. **c** Band gap of different modulation phases with $m = 0.18a$ and the band diagram of supercell, the band gap can be gradually opened at the Dirac point, as $\theta_r = 2\pi/3$. **d** Illustration of the Kekulé modulation phase distribution in the designed topological cavity. **e** Simulated near-electric field of the SMDC TL. **f** Simulated far-electric field of the SMDC TL.

shown in Fig. 2a. Kekulé modulation is introduced by constructing a continuous phase change from 0 to $2\pi$ around the cavity center to introduce a $2\pi$ vortex bandgap. The simplified formula (see Supplementary Materials S1 for details) of the displacement vector **m** can be written as follows:

$$\mathbf{m} = m\,e^{i(\theta_0 - \theta_r)} \tag{1}$$

where $m$ is the magnitude of displacement, $\theta_0$ is the initial relative phase of the three dotted circles in Fig. 2a, and $\theta_r = a\tan\left(\frac{y}{x}\right)$ represents the angle between the center of the primitive cell and the center of the cavity, as illustrated in Fig. 2b. Due to the double-metal waveguide structure, the effective refractive index contrast is sufficient to produce the needed band gap (see Supplementary Materials Fig. S2 for details). For our design, a calculated band gap width of 3% is obtained with $m = 0.18a$, as shown in Fig. 2c. The bandgap at the Γ point was approximated as the bandgap for our design[19]. Since the losses of the modes far from the Γ point (origin from the bulk band) are much larger than those of the topologically protected mode near the Γ point (topological mid-gap mode), these modes do not lase and have little effect on the topological mode. Therefore, for the topological mode operation of our device, the bandgap at the Γ point is the critical parameter. A large $m$ can result in a larger bandgap and out-of-plane emission due to the increasing coupling strength of the second-order grating. However, a too-large $m$ can cause lattice deformation and thus device degradation. Therefore, $m$ needs to be optimized, and devices with different $m$ values are constructed in this study.

Due to the position shifts of the spaced sites in each supercell, the original $C_6$ symmetry of the honeycomb lattice is broken. However, a relatively high $C_{3V}$ symmetry of the vortex cavity is still maintained when considering the center of the right circle with no position shift as the nominal central position of the lattice, as shown in Fig. 2b. The corresponding phase distribution of the cavity is illustrated in Fig. 2d,

showing the $C_{3V}$ symmetry of the whole cavity. Through 3D full-wave simulation, $C_{3V}$ symmetric near- and far-electric fields are obtained, as demonstrated in Fig. 2e, f. This $C_{3V}$-symmetric far field is the typical feature[18,19] of the topological zero lasing mode introduced by a topological cavity with $C_{3V}$ symmetry. Therefore, although the band gap width of the SMDC design is smaller than those of topological cavities with deep etching active region design[21], the 2D topological defect lasing mode, i.e., the Jackiw–Rossi zero mode, is also allowed. This mode is the only preferential lasing mode that has a much larger Q factor value than the other competing modes. Therefore, the SMDC devices tend to undergo single-mode operation naturally. Furthermore, this topological cavity mode has the advantage of achieving a large free spectral range (FSR), which is important for stable single-mode operation. For our device, the FRS is somewhat smaller than the conventional topological device with graphically etched active regions due to the smaller band gap. However, an FSR of 0.03 THz is obtained for our device, which ensues the stable single-mode operation over large device areas.

The electric-pumped SMDC TLs are fabricated based on a THz QCL wafer with a gain from 3.0 THz to 3.6 THz. To match this emission wavelength range, topological cavities with proper lattice constants are designed and fabricated. The fabrication of SMDC TLs is based on traditional double-metal waveguide device processing[27], including the patterning of the surface metal by optical lithography and lift-off (Methods). To verify the robustness of the SMDC design, devices with various lattice constants $a_0$, cavity parameters $m$ and boundary conditions were fabricated. In addition, second-order DFB devices from the same wafer were also fabricated for comparison.

## Experimental results

The measured lasing spectra of the SMDC TLs with different lattice constants $a_0$ and displacement magnitude $m = 0.18a$ are shown in Fig. 3a. Robust single-mode operations are obtained for devices within the entire current dynamic range and lasing frequencies from 3.32 to

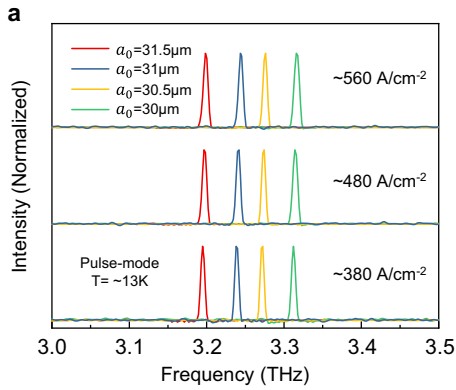

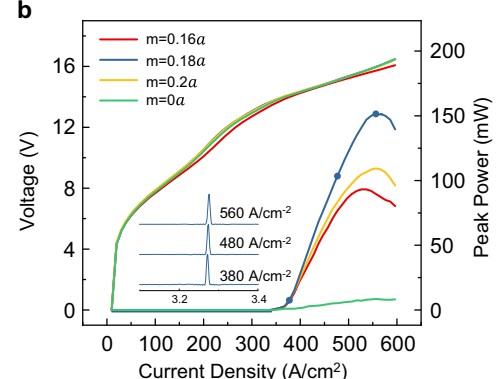

**Fig. 3 | Lasing spectra and L-I-V results of the SMDC TLs with different cavity parameters. a** Lasing spectra of devices with different lattice constants and $m = 0.18a$ under different injection current densities. **b** L-I-V curves of devices with different $m$ and the same lattice constant of $a_0 = 30.5\mu m$. Inset: Lasing spectra of the device with $m = 0.18a$ corresponding to injection currents labeled in the L-I-V curve.

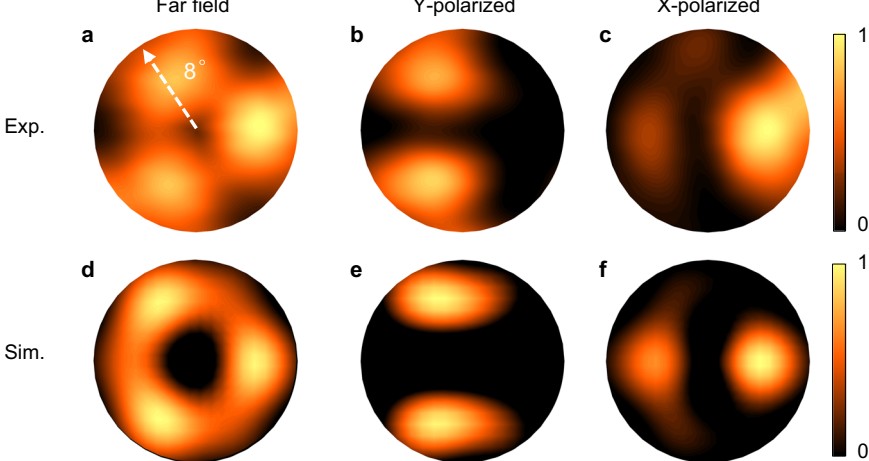

**Fig. 4 | Experimental and simulated far-field results. a–c** Experimental far-field and polarized far-field. For the far-field test, the operating temperature are set at 13 K, and the devices are driven near the peak power with 1 μs current pulses and a 1% duty cycle. **d–f** Simulated far-field and polarized far-field.

3.18 THz with changing lattice constants $a_0$. In addition, single-mode operations are also maintained at different operating temperatures, as detailed in Supplementary Materials Fig. S2. All these results show the robustness of the SMDC design for the excitation of the desired topological mode. Furthermore, the absorption boundary design has played an important role in the attainment of highly robust single-mode lasing (see Supplementary Materials Fig. S3). For comparison, the lasing spectra of the second-order DFB and trivial photonic crystal devices are also measured and shown in Supplementary Materials Fig. S4. The device with $m = 0a$ demonstrates multimode operation due to the unoptimized design of this photonic crystal structure. However, from another perspective, this result demonstrates the important role of the introduction of the vortex cavity for single-mode operation. For the second-order DFB device, although single-mode operation is demonstrated, the long far-field strip will cause considerable inconvenience for practical applications.

Next, the light-current-voltage (L-I-V) curves of the THz SMDC TLs with different values of $m$ and the same lattice constant $a_0 = 30.5\mu m$ are measured and depicted in Fig. 3b. The THz SMDC TLs demonstrate a maximal directly measured peak power of 150 mW with $m = 0.18a$. This output power increases by an order of magnitude considering the same device size compared with that of electrically pumped TLs with a deep-etched active region design[6,21,22]. Furthermore, this value is comparable to the best results obtained for other general types of THz surface-

emitting single-chip QCLs[24,25]. Through the three-dimensional (3D) full-wave finite element method simulation, the vertical radiative efficiency, defined as $\eta_{rad} = \alpha_\perp / \alpha_{total}$, is approximately 47.4% for the device with $m = 0.18a$ and $a_0 = 30.5\mu m$ (see Supplementary Materials S6 for details). Compared with typical THz surface-emitting single-chip QCLs with other physics mechanisms reported previously, our device demonstrates fairly competitive vertical radiation efficiency, as shown in Table S1. Therefore, enough gain provided by the surface metal cavity design and large vertical radiation loss and efficiency introduced by the topological photonics design bring in the increase of the output power of our device together. Additionally, the output power can be increased by increasing the device size, as shown in Supplementary Materials Fig. S7; this provides the potential for higher power.

To visualize the emitting beam profile, far-field patterns of the SMDC TLs are measured using a customized experimental setup with a high-sensitivity Golay cell detector scanning the curve of a sphere. The testing optical path diagram is provided in Supplementary Materials Fig. S8 in detail. As shown in Fig. 4, the far-field pattern of the THz SMDC TL exhibits $C_{3v}$ symmetry, which is consistent with the theoretical results of the Jackiw–Rossi zero mode. The far-field patterns under different polarization test conditions obtained by using a linear polarizer to filter the cross-polarized components are also consistent with the theoretical results of vector-polarized beams; these results further indicate the lasing of the designed topological zero mode.

## Far-field patterns tuning of the SMDC TLs

For practical applications, the control of far-field beam pattern[28] is also an important issue along with the output power. The far-field pattern of the topological lasing mode is related to the phase distribution of the topological cavity. Therefore, through phase distribution modulation, various far-field patterns can be obtained. The vortex phase modulation function can be written as follows:

$$\theta'_r = \pi \left( \frac{2 Arcsin\left(\frac{\theta_r}{\pi}\right)}{\pi} \right)^q \tag{2}$$

where $q$ is the modulation parameter. The phase modulation function and phase distribution for the cavity with phase modulation parameter $q$ of 3 are shown in Fig. 5a, b, respectively. Figure 5c, d shows the corresponding theoretical and experimental far-field patterns, demonstrating the conversion from the original $C_{3v}$ pattern to the double-lobed pattern. For $q = 1.5$, the tuning effect is also achieved, and a doughnut pattern is obtained, as shown in Fig. 5g, h. Furthermore, the far field maintains the characteristics of vector polarization, and the lasing spectral properties also remain unchanged (details in Supplementary Materials Fig. S9). Therefore, this SMDC design provides an effective and convenient approach to tune the far-field pattern through topological photonics due to the simplified device process compared with traditional designs with etched active regions.

## Discussion

In summary, we demonstrated high-power electrically pumped THz TLs based on the SMDC design. The devices lased robustly in the Jackiw–Rossi zero mode, which are exemplified from the stable single-mode lasing spectra and far-field patterns. Due to the non-destructive active region, the THz SMDC TLs demonstrate a maximal peak power of 150 mW; this value is an order of magnitude larger than that of previously reported electrically pumped THz TLs considering the same size. This work overcomes the strong limitations of conventionally deep etching topological cavity design for increasing the output power of electrically pumped TLs. In addition, adjustable far-field patterns are obtained by introducing a nonuniform phase distribution while keeping the spectral and vector-polarization characteristics unchanged. Our study provides an effective and convenient approach to tune the far-field pattern through topological photonics due to the simplified device process of our devices. Here, the scheme of the surface topological cavity facilitates an approach for accessing high-performance electrically pumped TLs. In principle, the surface topological cavity design will also applicable to semiconductor lasers at other wavelength ranges, such as near-infrared and mid-infrared wavelengths. These results may greatly promote the further development and practical application of electrically pumped TLs.

## Method

### Device fabrication

The THz QCL chip is based on a double-well GaAs/Al$_{0.15}$Ga$_{0.85}$As design with the gain bandwidth ranging from 3.0 to 3.6 THz. The fabrication of the THz SMDC TLs starts from the double-metal waveguide device processing of In-Au thermocompression wafer bonding. The semi-insulating GaAs substrate of the original wafer was then removed by lapping and selective wet etching. Next, the positive-resist lithography process was applied to etch away the 150 nm thick highly doped GaAs contact layer under the topological cavities using H$_3$PO$_4$:H$_2$O$_2$:H$_2$O etchant with 1:1:10 concentration. The circular air holes were then defined by image-reversal lithography followed by a sequence of Ti/Au (40/300 nm) metal deposition and lift-off as top metallic layers. To introduce the absorption boundary, the highly-doped absorption layer in the region with width of 15 μm along the edge of the topological cavity was retained. Then, silicon oxide with thickness of 1 μm was grown as the hard mask to etch the active region. Finally, hexagonal mesa structures were etched down to the bottom metal layer to avoid lateral current spreading. Three spaced rectangular electrodes with a size of 100*100 μm were introduced at the hexagonal boundary of the devices for wire bonding.

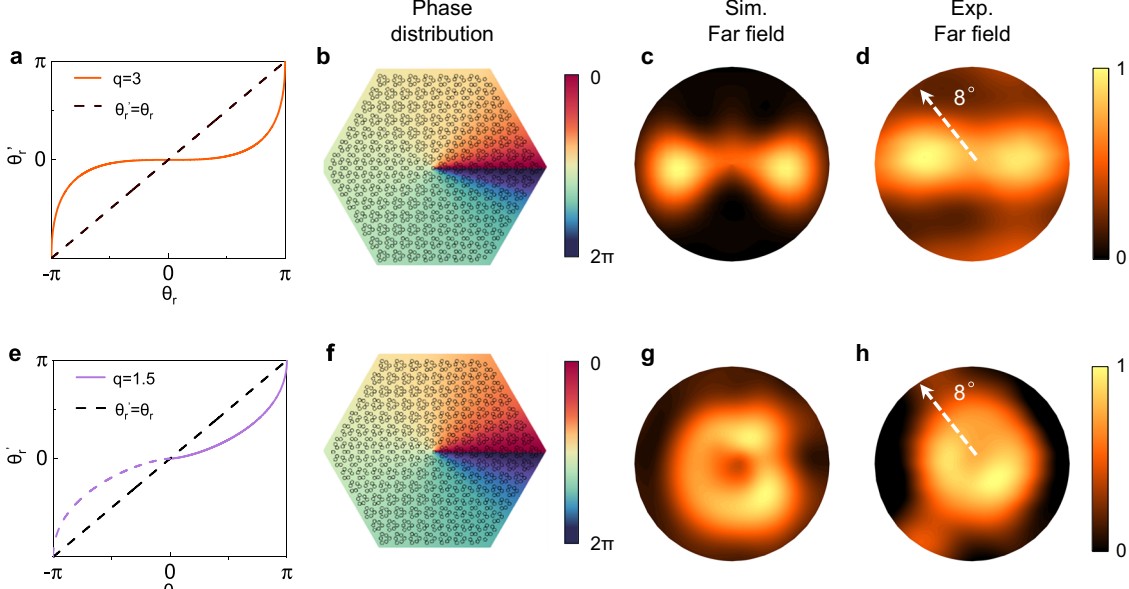

**Fig. 5 | Far-field symmetry tuning through phase distribution modulation.** **a**, **b** Phase distribution with a phase modulation parameter $q$ of 3. **c**, **d** Simulated and experimental far-field patterns with a phase modulation parameter $q$ of 3. **e**, **f** Phase distribution with a phase modulation parameter $q$ of 1.5. due to the presence of the mirror symmetry axis, the purple dotted line is partially symmetrical from the solid line. **g**, **h** Simulated and experimental far-field patterns with phase modulation parameter $q$ of 1.5. For the far-field test, the operating temperature is set at 13 K, and the devices are driven near the peak power with 1 μs current pulses and a 1% duty cycle.

## Characterization

For the pulsed mode light-current-voltage (L-I-V) and spectral characteristics measurements, a 1-µs-long pulse with 10 kHz signal cycle (1% duty-cycle) was chosen to drive the various devices at 13 K under different injection currents. The output power of the lasers was directly measured by Thomas Keating (TK) terahertz absolute power meter without any corrections or focusing optics. The lasing spectra were measured using a Fourier transform infrared spectrometer (Bruker; VERTEX 80 v) with a DTGS detector and a resolution of 0.2 cm$^{-1}$. For the far-field test, the operating temperature was set at 13 K, and the devices were driven near the peak power with 1 µs current pulses and a 1% duty cycle. The far-field patterns were measured with a high-sensitivity Golay cell detector scanning on the curve of a sphere with a radius about 15 cm. The high-frequency driven pulses were modulated into low frequency (20 Hz) envelopes by the signal generator, and the light intensity detected by the Golay cell detector was characterized by the voltage output of a lock-in amplifier.

## Simulation

All of the 3D full-wave and 2D TM-wave simulations were carried out based on the finite element method. In the simulation, the effective refractive index of the active region was 3.6, and the thickness of the active region was 11.7 µm. The refractive index of Au was $248 + 1323i$ and the thickness of the upper Au layer was 300 nm. The refractive index of the highly doped contact layer serving as the absorbing boundary was calculated to be $4.69 + 20.25i$ through the Drude–Lorenz model. The air-hole pattern was generated with the Layout Editor Software and imported into the simulation software.

## Reporting summary

Further information on research design is available in the Nature Portfolio Reporting Summary linked to this article.

## Data availability

Data supporting key conclusions of this work are included within the article and Supplementary information. All raw data that support the findings of this study are available from the corresponding authors upon request.

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

## Acknowledgements

This work is supported by National Natural Science Foundation of China (Grant Nos. 62222408, Grant Nos.12274404, Grant Nos. 62274014, Grant Nos. 62235016), S.Q.Z. acknowledges support from National Key Research and Development Program of China (2021YFB3201900) and Youth Innovation Promotion Association of the Chinese Academy of Sciences (2022112); S.M.L. and Q.Y.L. acknowledge support from Beijing Municipal Science & Technology Commission (Z221100002722018). The authors would like to thank Ping Liang and Ying Hu for their help with device processing.

## Author contributions

J.H.L. conceived the idea of the research, J.H.L. and Y.F.X. performed the theoretical and numerical calculations. Y.F.X. carried on the sample fabrications. J.H.L., S.Q.Z., Q.Y.L., and Y.F.X. wrote the paper. K.Y.X. and Y.Q.S. participated in schematic drawing of figures. R.S.L., Q.Y.L., J.C.Z., S.M.L., S.Q.Z. F.M.C., F.Q.L., and J.Q.L. discussed the results, L.J.W. and N.Z. performed QCL wafer growth. J.H.L. and Y.F.X. carried out spectral analysis; L-I-V and far-field measurements. S.Q.Z., F.Q.L., and J.C.Z. supervised the project.

## Competing interests

The authors declare no competing interests.
