## [Peer Review File · Nature Communications]

High-power electrically pumped terahertz topological laser based on a surface metallic Dirac-vortex cavityREVIEWER COMMENTS

Reviewer #1 (Remarks to the Author):

The authors report a THz quantum cascade laser (QCL) with a topological cavity based on a Jackiw-Rossi zero mode. The authors particularly focused on the increase of the output power under single mode operation, which is critical for practical use, and thus employed a surface metal design that does not directly pattern the active region buried in the semiconductor layer structure of the QCL. They applied a reported strategy of designing a topological zero mode [referenced as [18,19]] to their surface metal structure and found a cavity resonance capable of single mode lasing. In contrast to previously reported electrically-driven QCLs with topological cavities, the authors observed high output power over 150 mW, which is, according to the authors, comparable to the best single-chip surface emitting THz QCL of similar size. The author's claims are well supported by the experimental data and numerical simulations. The manuscript is overall well written, although there are occasional typos and logical flaws. However, I do not understand how the topological cavity increases the output power. If untouching the active region predominantly contributed to the increase of the output power, the topological design can be regarded as merely a design for single mode lasing, which can be achieved by many other design strategies [e.g. Nature 618 727 (2023)]. In my opinion, the current work needs critical assessment on how the fusion between the surface metal and the topological design results in the increase of output power. The comparison with DFB and simple photonic crystal devices provided in the supplementary material is not convincing. These control samples were not optimized for high output power and cannot explain why the topological design is suitable for high output power. For these reasons, I recommend a major revision of the manuscript before being accepted for publication. Other comments are provided below.

- (1) In the introduction, the authors claimed that the surface metal design is applicable for near-infrared. I disagree with this point since metal absorption is huge in such a wavelength range.
- (2) Some airholes in the metal photonic crystal are connected to each other. How detrimental is this for the operation of the laser?
- (3) The band structure in Fig. 2c contains two sets of lines. What are the origins of them and how did the author determine the bandgap irrespective of the presence of the dispersion crossing the gap?
- (4) The field distribution shown in Figure 2e suggests very weak localization of the cavity mode. I understand that weak confinement is advantageous for eliminating the presence of other confined modes. But such an approach is possible with other designs like wavelength-shifted DFB lasers. How the topological design is superior to the others? This type of discussion has been done in ref. [18] but the authors should elaborate on it for their own surface metal structure.
- (5) Considering the field distribution in Fig. 2e, loss of the cavity mode could be governed by the edge absorption, rather than the vertical radiation, which will be detrimental for high output power. Can the authors estimate the loss budget of the laser?
- (6) Related to comment (5), threshold gain difference among the optical modes could be determined predominantly by the edge absorption in the device. In such a case, how meaningful to use the topological cavity mode for single mode lasing?
- (7) In the supplementary material S5, the authors mentioned that "the instability of the normal band edge mode" results in multimode lasing. I do not understand this. It is merely related to the presence of multiple modes and their closeness of required gain for lasing. Even using defect-less designs of photonic crystals, it is possible to have a large-threshold gain difference without using a topological mode [e.g. Nature 618 727 (2023)]. I do not think that the comparison with the careless-designed photonic crystal band-edge structure is enough to claim the superiority of the topological cavity mode.

Reviewer #2 (Remarks to the Author):

The authors reported experimental results on electrically pumped terahertz lasers, using a novel topological cavity. The results are very interesting for the community and I recommend it for publication, once the following questions are answered.

1) It is not appropriate to have "High-power" in the title, because the power is not high among QCLs.

Wang Z, Liang Y, Meng B, et al. Large area photonic crystal quantum cascade laser with 5 W surface-emitting power[J]. Optics Express, 2019, 27(16): 22708-22716.

2) Since there have been many different types of topological cavities proposed, it might be better to specify "Dirac-vortex" cavity in the paper's title to be clear.

3) Why are the two contacts made on the same side? The carrier injection will be non-uniform. Is there data showing the carrier distribution in the cavity? Can the contacts be on opposite sides of the wafer?

4) Can the device be made much larger than the current size of 1mm?

5) The surface metal reflects and prevents the light from being vertically emitted. Is there an estimation of the out-coupling efficiency of the design?

6) In Fig.1b, the air holes in the cavity center are connected in pairs in the SEM images, different from the air hole at the boundary of the cavity. Can the authors explain why?

7) Please include pumping conditions of far-field measurements in Fig.4 and Fig.5.

Reply to Referee #1

Original general comment:

The authors report a THz quantum cascade laser (QCL) with a topological cavity based on a Jackiw-Rossi zero mode. The authors particularly focused on the increase of the output power under single mode operation, which is critical for practical use, and thus employed a surface metal design that does not directly pattern the active region buried in the semiconductor layer structure of the QCL. They applied a reported strategy of designing a topological zero mode [referenced as [18,19]] to their surface metal structure and found a cavity resonance capable of single mode lasing. In contrast to previously reported electrically-driven QCLs with topological cavities, the authors observed high output power over 150 mW, which is, according to the authors, comparable to the best single-chip surface emitting THz QCL of similar size. The author's claims are well supported by the experimental data and numerical simulations. The manuscript is overall well written, although there are occasional typos and logical flaws. However, I do not understand how the topological cavity increases the output power. If untouching the active region predominantly contributed to the increase of the output power, the topological design can be regarded as merely a design for single mode lasing, which can be achieved by many other design strategies [e.g. Nature 618 727 (2023)]. In my opinion, the current work needs critical assessment on how the fusion between the surface metal and the topological design results in the increase of output power. The comparison with DFB and simple photonic crystal devices provided in the supplementary material is not convincing. These control samples were not optimized for high output power and cannot explain why the topological design is suitable for high output power. For these reasons, I recommend a major revision of the manuscript before being accepted for publication. Other comments are provided below.

Our reply:

We sincerely thank the referee for the positive comments on this work. And we are very much grateful for the insightful comments and suggestions that help us to improve the quality of this work. We included detailed interpretations and elaborated the physical mechanism for high output power in the revised version to address the referee's

comments. Detailed discussion will be given in the replies below.

In this work, we focused on the bottleneck issue of low output power for the electrically pumped THz semiconductor topological lasers (TLs) which have been demonstrated with excellent mode robustness against optical perturbations or defects. We proposed a novel surface metallic topological cavity by defining the patterns into the top metal layer leaving the active region intact for robust single mode and high power operation simultaneously. This is strikingly different from all the previous demonstrated electrically pumped THz TLs which rely on disruptive patterning into the active region for robust single mode operation while give little consideration to high power output. The surface topological cavity design untouched the active region significantly contributed to the higher output power compared to the traditional electrically pumped topological lasers, and leaved great potential for the increase of output power. Therefore, this work not merely realized stable single topological mode lasing, but solved the bottleneck problem of low output power of topological electrically pumped semiconductor lasers, which may greatly promote the further development and practical applications of electrically pumped TLs

The high output power performance of our device is further corroborated by the high radiative (out-coupling) efficiency due to the topological photonic design. We performed 3D Finite element simulation and calculated the radiation loss and efficiency of our topological cavity lasers in the revised manuscript. Based on the simulation results in Fig. 2e, the radiation efficiency (defined as $\eta_{rad} = \alpha_{\perp}/\alpha_{total}$) about 47.4% for the device with with $m = 0.18a$ and $a_0 = 30.5\mu\text{m}$ is obtained for the topological mode. This relatively high out-coupling efficiency originates from the coherent constructive emission of the topological mode over the entire lattice area. Table 1 compared our results with other typical high-power surface-emitting THz QCL reported previously. Clearly, our device exhibits fairly competitive radiation loss and efficiency compared with devices based on different photonic designs, providing the huge potential for high output power. For the absolute output power of these devices, the epitaxial material used in the device fabrications is another critical factor, which may introduce the inconsistency between the radiation efficiency and output power. Therefore, enough gain provided by the surface metal cavity design and large vertical radiation loss and efficiency introduced by the topological photonics design result in the increase of the output power of our device together.

Table 1. Radiation loss and efficiency for different single-chip THz surface emitting QCLs

Work	Physics mechanism	Radiation loss	Radiation efficiency η_{rad}
Nat. Commun. 3, 952, 2012	Graded gratings	11cm^{-1}	$\approx 34\%$
Nat. Commun. 9, 1049, 2018	Hybrid second-and fourth-order gratings	6cm^{-1}	$\approx 30\%$ ⁽¹⁾
Nat. Commun. 5, 5884, 2014	Photonic quasi-crystal	$2.7\text{cm}^{-1(2)}$	$\approx 16\%$
Current work	Topological mid-gap mode	17.5cm^{-1}	$\approx 47.4\%$

(1) Estimated with the typical parameters from (*J. Appl. Phys.* 97, 053106, 2005; *Appl. Phys. Lett.* 100, 261111, 2012; *Appl. Phys. Lett.* 105, 181118, 2014).

(2) Calculated from the Q factor.

In fact, our work demonstrated one feasible technical solution for stable single operation with high output power of semiconductor lasers. There are also many other design strategies for high performance and each of them has the potential to demonstrate superior performances based on creative designs, such as the results in [*Nat. Commun.* 9, 1407, 2018] and [*Nature* 618, 727, 2023]. As the referee pointed out, the comparison with DFB and simple photonic crystal devices provided in the supplementary material is not convincing, since these control samples were not optimized for high output power. In the revised manuscript, we remove the output power comparison with the DFB and simple photonic crystal devices. Instead, we have added the theoretical calculation results and comparisons with other typical high-power surface-emitting THz QCLs reported previously in the revised supplementary materials S6 to demonstrate the potential for high output power of our device. Our work demonstrates one feasible approach to significantly increase the output power of electrically pumped topological semiconductor lasers and would be an important result in the research community of electrically pumped semiconductor lasers.

In further, we have also checked the manuscript carefully to revise the occasional typos and logical flaws. And we thank the referee again for the comments, which helps us to improve the quality and clarity of the presentation in the revised manuscript.

Original comment (1):

In the introduction, the authors claimed that the surface metal design is applicable for near-infrared. I disagree with this point since metal absorption is huge in such a

wavelength range.

Our reply:

We thank the referee for this comment. We totally agree with the referee that the “metal absorption is huge in such a wavelength range” which makes the surface metallic waveguide design not applicable in the near infrared range. The statement in the previous main text may be not clear. What we wanted to claim is that the “surface topological cavity design” that fabricated in the surface metal or waveguide layers without etching the active region could also be applicable to electrically pumped semiconductor lasers in other wavelength ranges. This approach is applicable when sufficient refractive index contrast is achieved, either by metallic cavity design in this work or by etching of waveguide layers in telecom bands (like *Light: Science & Applications* 8, 108, 2019, and *Nat. Photon.* 16, 279-283, 2022). We revised the statement in the introduction in the revised manuscript.

Original comment (2):

Some air holes in the metal photonic crystal are connected to each other. How detrimental is this for the operation of the laser?

Our reply:

We thank the referee for raising this point. We have added the explanation for the connections of some holes, and given the evaluation of the effect of these connections on the device performance in the revised manuscript.

The connected air holes in some areas are not going to affect the laser performance significantly for the following two reasons.

Firstly, the connecting areas of the air holes are rather small compared with the area of the unit cell and won't affect the general symmetry of the unit cell. As is demonstrated in Figure R1, no significant change on the modal distribution is observed due to the connections according the simulation results. In Figure R1, L is the width of the connecting region, and r_0 is the radius of the air hole. We simulated the proportion of the energy flow density in the air hole region after considering the connecting airholes parameter with L/r_0 from 0 to 1, which shows a nearly unchanged modal distribution regarding to the varying L/r_0 ratio.

Secondly, the connection of the air holes is only observed in certain area of the cavity with larger values of m for higher coupling strength, where the airholes are closely packed resulting from the imperfect metal patterns during liftoff process. This area accounts for about 16% (25%) of the total device area with $m = 0.16a$ ($m = 0.18a$). Combined with the previous calculation of the weak effect of the connecting airholes on the modal distribution, we are sure that the connected airholes in some areas will have a negligible influence to the device performance. This has been verified through the well agreement between the experimental and theoretical far-field results.

Figure R1. Simulation of the proportion of the energy flow density in the air hole region with different connecting width, the insert pictures show the modal distribution with $L/r_0 = 0; \frac{1}{3}; \frac{2}{3}; 1$ respectively.

Original comment (3):

The band structure in Fig. 2c contains two sets of lines. What are the origins of them and how did the author determine the bandgap irrespective of the presence of the dispersion crossing the gap?

Our reply:

We thank the referee for this comment. The two sets of lines in the band diagram in Fig. 2c origin from the honeycomb lattice consisting a hexagonal supercell composed of six neighboring sites as unit cell. For this honeycomb lattice, the two Dirac points from the Brillouin-zone boundary ($\pm K$ points below the light cone) are folded to the zone center (Γ point above the light cone), forming a four-by-four double Dirac cone dispersion. By shifting three spaced sites along a specific displacement vector \mathbf{m} in the honeycomb sublattice, the band gap around Γ point opens for all values of θ_r . The right picture in Fig. 2c is the band diagram of the TM mode with parameters of $m = 0.18a$ and $\theta_r = 2\pi/3$, which corresponds to the minimum band gap value for different θ_r .

Indeed, absolute photonic bandgap across the major directions in the Brillouin zone is difficult to obtain due to the presence of the dispersion crossing the gap. This is a universal characteristic for the structure of weak refractive index difference design. To determine the bandgap, we have used a similar method reported in [*Nat. Photon.* 16,279-283, 2022]. Since the losses of the modes far away from Γ point (origin from the bulk band) are much larger than that of the topological protected mode near Γ point (topological mid-gap mode), these modes would not lase and have little effect on the topological mode. Therefore, for the topological mode operation of our device, the bandgap value at Γ point is a critical parameter and using this value to represent the bandgap of our design is reasonable. We have added this explanation in the revised manuscript.

Original comment (4):

The field distribution shown in Figure 2e suggests very weak localization of the cavity mode. I understand that weak confinement is advantageous for eliminating the presence of other confined modes. But such an approach is possible with other designs like wavelength-shifted DFB lasers. How the topological design is superior to the others? This type of discussion has been done in ref. [18] but the authors should elaborate on it for their own surface metal structure.

Our reply:

We thank the referee for raising this valuable point. We have added the elaboration of the advantages of our topological design over the other designs in the revised manuscript.

Compared with other designs, our device operated in the robust 2D topological defect lasing mode, that is the Jackiw-Rossi zero mode. The topological photonic design opened a photonic bandgap and created this topologically protected mid-gap mode, as shown in Fig. 2. This is the only one preferential lasing mode, which has much larger Q factor value than other competing modes. Therefore, topological photonic devices tend to single mode operation naturally. In further, this topological cavity mode has the advantage for realizing a large free spectral range (FSR), which is important for stabler single-mode operation. This large FSR originates from the construction of the single cavity mode at the middle of the Dirac spectrum where the optical density of other states vanishes, so that the FSR is spectrally non-uniform and peaks at the Dirac frequency [*Nat. Nanotech.*, 15, 1012–1018, 2020]. However, it would be impossible for other photonics designs to realize such mid-gap lasing mode. For our device, the FRS is somewhat smaller than the conventional topological device with graphically etched active regions due to the smaller band gap. However, a FSR of 0.03 THz is obtained for our device, which ensues the stable single-mode operation over large device areas. We have added these statements in the revised manuscript.

Original comment (5):

Considering the field distribution in Fig. 2e, loss of the cavity mode could be governed by the edge absorption, rather than the vertical radiation, which will be detrimental for high output power. Can the authors estimate the loss budget of the laser?

Our reply:

We thank the referee for this comment. We have added the simulation results of the loss of the cavity mode in the revised manuscript to estimate the loss budget of our device. Through the calculation, the radiation efficiency of the laser (defined as $\eta_{rad} = \alpha_{\perp} / \alpha_{total}$) about 47.4% is obtained for the device with $m = 0.18a$ and $a_0 = 30.5\mu\text{m}$, which is beneficial for the high output power of the device. We have added the calculation details in revised supplementary materials S6.

To evaluate the loss budget of the laser, we have calculated the loss of the cavity mode using a three-dimensional (3D) full-wave finite element method. Firstly, we have calculated Q factors for different modes, and results are shown in Figure R2. The photon

loss rate γ_r due to the vertical radiation of the laser is estimated by extracting the time-averaged integrated power flow through the open air hole domains and normalizing it with respect to the resonator energy in the 3D simulation (*Supplementary of [Nat. Commun. 5,5884,2014]*):

$$\gamma_r = \frac{\Phi}{E_{res}} = \frac{\int_A (E \times H) \cdot \hat{n} dS}{\int_V (\epsilon |E|^2 + |H|^2 / \mu) dV} \quad (R. 1)$$

Where E and H represent the electric and magnetic fields, respectively, \hat{n} is the unit normal vector of the circle air domains, ϵ the dielectric constant and μ the permeability. The corresponding quality factors have been derived from the relation $Q_{vertical} = v/\gamma_r$, where v is the eigenfrequency of the topological mode. And the radiative out-coupling efficiency η_r is assumed to be proportional to $\frac{Q_{total}}{Q_{vertical}}$ where:

$$Q_{total} = \left(\frac{1}{Q_{inplane}} + \frac{1}{Q_{vertical}} \right)^{-1} \quad (R. 2)$$

Based on the 3D simulation, a photon loss rate of $\gamma_r \approx 28.5$ GHz from integral results through equation (R.1) is obtained, which corresponds to $Q_{vertical} = 140$. Considering the calculated $Q_{total} = 96$ from the 3D simulation, an in-plane quality factor $Q_{inplane} = 320$ is obtained. As a result, an out-coupling efficiency of $\eta_r \approx 68.5\%$ is estimated. This result is obtained without considering the waveguide loss.

Figure R2. COMSOL 3D simulation of the Q factor

Next, to obtain the vertical radiation efficiency, we use the experimental results of the F-P device from the same wafer as the topological cavity device to modify the

theoretical calculation considering the waveguide loss. For the F-P device with 2 mm cavity length and 100 μm ridge width, we have $\alpha_m = \frac{1}{2L} \ln\left(\frac{1}{R_1 R_2}\right) = \frac{1}{L} \ln\left(\frac{1}{R}\right)$, and the facet reflectivity for metal-metal waveguide structure with waveguide width of 100 μm and active region thickness of 11.7 μm around 3.2 THz is about 80% (*J. Appl. Phys.* 97,053106,2005). Therefore we obtain $\alpha_m \approx 1.1\text{cm}^{-1}$. Meanwhile, the total optical loss coefficient α_{total} for F-P plasmonic QCLs operating in the wavelength range of 3~4THz has been experimentally measured in the range of 10~15 cm^{-1} (*Appl. Phys.Lett.*100, 261111,2012; *Appl. Phys.Lett.*105,181118,2014). Considering $\alpha_{total} = \alpha_m + \alpha_w$, where α_w is waveguide loss, we take the $\alpha_{total} = 12.5\text{cm}^{-1}$ and $\alpha_w = 11.4\text{cm}^{-1}$ is obtained.

Figure R3. L-I-V curves of the F-P device with 2 mm cavity length and 100 μm ridge width.

Figure R3 shows the L-I-V curves of the F-P device with slope efficiency of 28 mW/A . In theory, the slope efficiency of the F-P device can be written as

$$\eta_{slope_FP} = \eta_i \frac{N\hbar\omega}{e} \eta_{rad_FP} = \eta_i \frac{N\hbar\omega}{e} \frac{\alpha_m}{\alpha_m + \alpha_w} \quad (\text{R. 3})$$

Meanwhile, as for the topological cavity device, the total optical loss coefficient and the slope efficiency can be written as

$$\alpha_{total} = \alpha_{\perp} + \alpha_{\parallel} + \alpha_w \quad (\text{R. 4})$$

$$\eta_{slope_Topo} = \eta_i \frac{N\hbar\omega}{e} \eta_{rad_Topo} = \eta_i \frac{N\hbar\omega}{e} \frac{\alpha_{\perp}}{\alpha_{\perp} + \alpha_{\parallel} + \alpha_w} \quad (\text{R. 5})$$

Where α_{\perp} represents optical loss coefficient due to the vertical radiation, α_{\parallel} is the in-plane mirror loss which can be obtained from $\alpha = \frac{2\pi * n_{eff}}{\lambda * Q}$. Hence α_{\perp} ($Q_{vertical}=140$)

and $\alpha_{\perp} + \alpha_{\parallel}(Q_{total}=96.7)$ are calculated to be 17.5cm^{-1} and 25.5cm^{-1} respectively. Therefore, after taking the waveguide loss α_w in to consideration, we obtain $\eta_{rad_Topo} = \frac{17.5}{(25.5+11.4)} = 0.474$ for our topological device with $m = 0.18a$ and $a_0 = 30.5\mu\text{m}$. The slope efficiency $\eta_{slope_Topo} = 151\text{mW/A}$ of this topological device is obtained from the P-I-V curve in Fig. 3b experimentally. Therefore, we have the theoretically calculated slope efficiency ratio for the topological and FP devices $\frac{\eta_{rad_Topo}}{\eta_{rad_FP}} = \frac{0.474}{1.1/12.5} = 5.386$, which is in excellent agreement with the experimental data $\frac{\eta_{slope_Topo}}{\eta_{slope_FP}} = \frac{151}{28} = 5.393$. This verifies the reliability of the numerical simulation.

This calculation shows that about 47.4% of the photon loss channels are vertical radiation when the waveguide loss is considered for the topological device with $m = 0.18a$ and $a_0 = 30.5\mu\text{m}$, which is much greater than the radiation efficiency of the conventional metal-metal F-P devices and ensures the high output power.

Original comment (6):

Related to comment (5), threshold gain difference among the optical modes could be determined predominantly by the edge abruption in the device. In such a case, how meaningful to use the topological cavity mode for single mode lasing?

Our reply:

We thank the referee for raising this question. For the topological mode operation of our device, the threshold gain difference is mainly obtained due to the modal loss difference between the topological mode and bulk modes, and a high threshold margin of $3\text{cm}^{-1}(Q_1=96.7 \& Q_2=87)$ is obtained, as the simulated results shown in Figure R2. This ensures stable single-mode operation of the device. For our device, we have added the absorbing edge, that is the $15\text{-}\mu\text{m}$ highly doped contact layer, to increase the losses of the whispering-gallery like modes and ensure the single mode operation of the topological mid-gap mode. This also increases the loss of the bulk modes due to the divergent field distribution, which is further beneficial for single mode stability.

For the topological cavity, the single mid-gap mode can lase first where the grating feedback is the strongest. For other photonic designs, such as photonic-crystal surface-emitting laser, there are at least two high-quality-factor (Q) band-edge modes

competing for lasing, which will affect the stability of single mode operation. To obtain stable single mode operation, complex photonic structure designs are required to realize mode selection. However, for the topological cavity, there is only one preferential lasing mode, which has a much larger Q factor value than its competing ones. Therefore, topological photonic devices tend to single mode operation naturally. In further, as described in reply to comment 4, the topological cavity mode realizes a large free spectral range (FSR), which offers the exciting opportunity to realize stabler single-mode operation over large areas.

Original comment (7):

In the supplementary material S5, the authors mentioned that “the instability of the normal band edge mode” results in multimode lasing. I do not understand this. It is merely related to the presence of multiple modes and their closeness of required gain for lasing. Even using defect-less designs of photonic crystals, it is possible to have a large-threshold gain difference without using a topological mode [e.g. Nature 618 727 (2023)]. I do not think that the comparison with the careless-designed photonic crystal band-edge structure is enough to claim the superiority of the topological cavity mode.

Our reply:

We thank the referee for this comment. In the supplementary material S5 in the original manuscript, we attributed the multi-mode operation of the device with $m = 0a$ to the instability of the normal band edge modes. As the referee pointed out, this statement is not rigorous. The multi-mode operation is attributed to the existence of many different modes with the similar gain condition for lasing in the band-edge structure due to the nonoptimized photonic design. We have modified the statement in the revised manuscript.

In the original manuscript, we actually wanted to use the experimental results of the device with $m = 0a$ to verify the effect of topology designs. With different values of m , different device performances were obtained. It is not a comparison between the topological lattice and normal photonic crystal lattice, but rather a comparison between topological designs with different parameters. Obviously, the carefully designed photonic crystals device also can realize a large-threshold gain difference (such as the

reference [*Nature* 618, 727, 2023]), or free-spectral range loss control in an open-Dirac cavity (such as the reference [*Nature* 608, 692, 2022]). Therefore, in the revised manuscript, we modify the statement and ascribe the multimode operating of the device with $m = 0a$ to an special case of topological design in the revised manuscript. In further, we have added the comparison of our device and typical high-power surface-emitting THz QCL reported previously and theoretical calculation results in the revised manuscript to display the high output power potential of our topology design.

Reply to Referee #2

Original general comment:

The authors reported experimental results on electrically pumped terahertz lasers, using a novel topological cavity. The results are very interesting for the community and I recommend it for publication, once the following questions are answered.

Our reply:

We thank the referee for the supportive and encouraging comments. Below, we will respond to the referee's specific comments one by one.

Original comment (1):

It is not appropriate to have "High-power" in the title, because the power is not high among QCLs.

Wang Z, Liang Y, Meng B, et al. Large area photonic crystal quantum cascade laser with 5 W surface-emitting power[J]. Optics Express, 2019, 27(16): 22708-22716.

Our reply:

We thank the referee for this comment. We used "High power" in the title for two reasons. For one thing, the output power value of our device is close to the best results of other general types of single-chip surface-emitting THz lasers with single mode operation. The reference in the comment is the result of the surface-emitting QCL device in the mid infrared wavelength range. Limited by the operating mechanism, output power of the THz device is much lower than mid- and far- infrared QCL device. In the revised manuscript, we compared the radiation efficiency of our device with other typical high-power surface-emitting THz QCL reported previously. Our device exhibits fairly competitive radiation efficiency, providing the huge potential for high output power. For another, the most important thing for our work is to address the bottleneck issue of power improvement of electrically pumped topological lasers. Through the novel design, our device demonstrates a maximum peak power of 150 mW, which is an order of magnitude improvement over that of previously reported electrically pumped THz TLs considering the same size, which may open up a new approach for high-performance electrically pumped TLs. In addition, in the revised manuscript, we have

added theoretical calculation results in the revised manuscript to display the potential for high output power of the topology design. Additionally, the output powers can be increased with increasing device sizes. Therefore, we think that the “high power” in the title is reasonable.

Table 1. Radiation loss and efficiency for different single-chip THz surface emitting QCLs

Work	Physics mechanism	Radiation loss	Radiation efficiency η_{rad}
Nat. Commun. 3, 952, 2012	Graded gratings	11cm^{-1}	$\approx 34\%$
Nat. Commun. 9,1049, 2018	Hybrid second-and fourth-order gratings	6cm^{-1}	$\approx 30\%$ ⁽¹⁾
Nat. Commun. 5, 5884, 2014	Photonic quasi-crystal	2.7cm^{-1} ⁽²⁾	$\approx 16\%$
Current work	Topological mid-gap mode	17.5cm^{-1}	$\approx 47.4\%$

(1) Estimated with the typical parameters from (*J. Appl. Phys.* 97, 053106, 2005; *Appl. Phys. Lett.* 100, 261111, 2012; *Appl. Phys. Lett.* 105, 181118, 2014).

(2) Calculated from the Q factor.

Original comment (2):

Since there have been many different types of topological cavities proposed, it might be better to specify "Dirac-vortex" cavity in the paper's title to be clear.

Our reply:

We thank the referee for providing this meaningful suggestion. To highlight the features of our structural design, we have revised our paper's title to “High-power electrically pumped terahertz topological laser based on surface metallic Dirac-vortex cavity”. We have also modified the corresponding statements and abbreviation in the revised manuscript.

Original comment (3):

Why are the two contacts made on the same side? The carrier injection will be non-uniform. Is there data showing the carrier distribution in the cavity? Can the contacts be on opposite sides of the wafer?

Our reply:

We thank the referee for raising this meaningful questions. We are sorry that we didn't communicate the information clearly enough in Fig. 1a in the original manuscript. In fact, the two contacts are made on different sides, the back metal contact was fabricated by electron beam evaporation on the n type GaAs substrate as negative electrode and three square-contact pads covered with metal layers were fabricated on the edges of the device as positive electrodes for uniform current injection. And we have corrected Fig. 1a to avoid misunderstandings as shown in the revised manuscript and we also emphasize this in the revised manuscript.

Unlike mid-IR QCLs, the entire top surface of the THz topological device except the air holes is metallized, and there shouldn't be any non-uniform current injection issues. The total area of the airholes is small compared with the whole device and the airhole regions can also be electrically pumped via lateral current spreading uniformly.

Original comment (4):

Can the device be made much larger than the current size of 1mm?

Our reply:

We thank the referee for this comment. Yes, the device can be made much larger than the current size of 1 mm. Due to the large free spectral range (FSR) of the topological mid-gap mode, stable single mode operation is expected with larger device size. In further, smaller divergence angle will be obtained with increasing device size [Nat. Photon. 16,279-283, 2022]. The biggest challenge for large-area device is the heat dissipation issue of the device, nevertheless, this issue could be potentially addressed by thinning the thickness of the active region accordingly. This will be our future work. We have added this statement in supplementary materials in the revised manuscript.

Original comment (5):

The surface metal reflects and prevents the light from being vertically emitted. Is there an estimation of the out-coupling efficiency of the design?

Our reply:

We thank the referee for raising this meaningful questions. We have added the

estimation of the out-coupling efficiency of our design in the revised manuscript.

We have calculated the out-coupling efficiency of the surface metallic topological cavity laser using a three-dimensional (3D) full-wave finite element method. The photon loss rate γ_r due to the vertical radiation of the laser is estimated by extracting from the 3D-simulated the time-averaged integrated power flow through the open air domains and normalizing it with respect to the resonator energy (*Supplementary of [Nat. Commun. 5,5884,2014]*):

$$\gamma_r = \frac{\Phi}{E_{res}} = \frac{\int_A (E \times H) \cdot \hat{n} dS}{\int_V (\epsilon|E|^2 + |H|^2/\mu) dV} \quad (R. 1)$$

Where E and H represent the electric and magnetic fields respectively, \hat{n} represent the unit normal vector of the circle air domains, ϵ the dielectric constant and μ the permeability. The corresponding quality factors have been derived from the relation $Q_{vertical} = v/\gamma_r$ where v is the eigenfrequency of the topological mode. And the radiative out-coupling efficiency η_r is assumed to be proportional to $\frac{Q_{total}}{Q_{vertical}}$ where:

$$Q_{total} = \left(\frac{1}{Q_{inplane}} + \frac{1}{Q_{vertical}} \right)^{-1} \quad (R. 2)$$

Here we get the value $\gamma_r \approx 28.5\text{GHz}$ from integral results through equation(R.1), which gives $Q_{vertical} = 140$ and from the 3D simulation we can obtained the $Q_{total} = 96.7$, which gives $Q_{inplane} = 320$ and $\eta_r \approx 68.5\%$. This result is obtained without considering the waveguide loss.

Figure R1. COMSOL 3D simulation of the Q factor

Next, we use the experimental results of the F-P device from the same wafer as the topological cavity device to modify the theoretical calculation considering the waveguide loss. For the F-P device with 2 mm cavity length and 100 μm ridge width, we have $\alpha_m = \frac{1}{2L} \ln\left(\frac{1}{R_1 R_2}\right) = \frac{1}{L} \ln\left(\frac{1}{R}\right)$, and the facet reflectivity for metal-metal waveguide structure with waveguide width of 100 μm and active region thickness of 11.7 μm around 3.2 THz is about 80% ([*J. Appl. Phys.* 97,053106,2005]). Therefore we obtain $\alpha_m \approx 1.1\text{cm}^{-1}$. Meanwhile, the total optical loss coefficient α_{total} for F-P plasmonic QCLs operating in the wavelength range of 3~ 4THz has been experimentally measured in the range of 10~15 cm^{-1} (*Appl. Phys.Lett.*100,261111,2012; *Appl. Phys.Lett.*105,181118,2014). Considering $\alpha_{total} = \alpha_m + \alpha_w$, where α_w is waveguide loss, we take the $\alpha_{total} = 12.5\text{cm}^{-1}$ and $\alpha_w = 11.4\text{cm}^{-1}$ is obtained.

Figure R2. L-I-V curves of the F-P device with 2 mm cavity length and 100 μm ridge width.

Figure R2 shows the L-I-V curves of the F-P device with slope efficiency of 28 mW/A . In theory, the slope efficiency of the F-P device can be written as

$$\eta_{slope_FP} = \eta_i \frac{N\hbar\omega}{e} \eta_{rad_FP} = \eta_i \frac{N\hbar\omega}{e} \frac{\alpha_m}{\alpha_m + \alpha_w} \quad (\text{R. 3})$$

Meanwhile, as for the topological cavity device, the total optical loss coefficient and the slope efficiency can be written as

$$\alpha_{total} = \alpha_{\perp} + \alpha_{\parallel} + \alpha_w \quad (\text{R. 4})$$

$$\eta_{slope_Topo} = \eta_i \frac{N\hbar\omega}{e} \eta_{rad_Topo} = \eta_i \frac{N\hbar\omega}{e} \frac{\alpha_{\perp}}{\alpha_{\perp} + \alpha_{\parallel} + \alpha_w} \quad (\text{R. 5})$$

Where α_{\perp} represent optical loss coefficient due to the vertical radiation, α_{\parallel} is the in-plane mirror loss which can be obtained from $\alpha = \frac{2\pi*n_{eff}}{\lambda*Q}$. Hence α_{\perp} ($Q_{vertical}=140$) and $\alpha_{\perp} + \alpha_{\parallel}$ ($Q_{total}=96.7$) are calculated to be $17.5cm^{-1}$ and $25.5cm^{-1}$ respectively. Therefore, after taking the waveguide loss α_w in to consideration, we obtain $\eta_{rad_Topo} = \frac{17.5}{(25.5+11.4)} = 0.474$ for the SMDC device with $m = 0.18a$ and $a_0 = 30.5\mu m$. Meanwhile, the slope efficiency η_{slope_Topo} of this SMDC device with $m = 0.18a$ is obtained to $151mW/A$ from the P-I-V curve in Fig. 3b experimentally. Therefore, we have the theoretically calculated slope efficiency ratio for the topological and FP devices $\frac{\eta_{rad_Topo}}{\eta_{rad_FP}} = \frac{0.474}{1.1/12.5} = 5.386$, which is in excellent agreement with the experimental data $\frac{\eta_{slope_Topo}}{\eta_{slope_FP}} = \frac{151}{28} = 5.393$. This verifies the reliability of the numerical simulation.

The out-coupling efficiency of the design about 47.4% is estimated, which is much greater than the radiation efficiency of the conventional metal-metal F-P device and ensures the high output power. We have added the calculation details of out-coupling efficiency of the device in revised supplementary materials S6.

Original comment (6):

In Fig.1b, the air holes in the cavity center are connected in pairs in the SEM images, different from the air hole at the boundary of the cavity. Can the authors explain why?

Our reply:

We thank the referee for raising this valuable point. In order to illustrate the problem clearly, we have added the explanation for the connections of some holes, and given the evaluation of the effect of this connections on the device performance in the revised manuscript.

The reason why some of the air holes are connected in pairs is that the metal between the air holes were stripped off in the metal stripping process due to the small distance between the holes. During device design, we shifted three spaced sites along a specific displacement vector \mathbf{m} in the hexagonal supercell to open the band gap. Larger shifts of the holes will introduce smaller distances between the adjacent holes, as is shown in Figure.R3, and affect the device fabrication process. Here we show the figure near the center of the cavity and the right boundary in Figure R3, which demonstrates the reason for these small distances between the adjacent holes clearly.

Figure.R3 Details of the distribution at different positions of the cavity. a) Illustration of the displacement within a single hexagonal supercell. b) Distribution near the center. c) Distribution near the right boundary. The gray color represents sites without displacement, and the circle sites with displacement are colored by their initial position.

In further, we have evaluated the effect of the fabrication imperfections on the device performance. The connected airholes in some areas will have a negligible influence to the device performance for two reasons.

Firstly, the connecting areas of the air holes are rather small compared with the area of the unit cell and won't affect the general symmetry of the unit cell. As is demonstrated in Figure R4, no significant change on the modal distribution is observed due to the connections according the simulation results. In Figure R4, L is the width of the connecting region, and r_0 is the radius of the air hole. We simulated the proportion of the energy flow density in the air hole region after considering the connecting airholes parameter with L/r_0 from 0 to 1, which shows a nearly unchanged modal distribution regarding to the varying L/r_0 ratio.

Secondly, the connection of the air holes is only observed in certain area of the cavity with larger values of m for higher coupling strength, where the airholes are closely packed resulting from the imperfect metal patterns during liftoff process. This

area accounts for about 16% (25%) of the total device area with $m = 0.16a$ ($m = 0.18a$). Combined with the previous calculation of the weak effect of the connecting airholes on the modal distribution, we are sure that the connected airholes in some areas will have a negligible influence to the device performance. This has been verified through the well agreement between the experimental and theoretical far-field results.

Figure R4. Simulation of the proportion of the energy flow density in the air hole region with different connecting width, the insert pictures show the modal distribution with $L/r_0 = 0; \frac{1}{3}; \frac{2}{3}; 1$ respectively.

In order to illustrate the problem clearly, we have added the explanation for the connections of some holes, and given evaluation of the effect of this process fabrication imperfections on the device performance in the revised manuscript.

Original comment (7):

Please include pumping conditions of far-field measurements in Fig.4 and Fig.5.

Our reply:

We thank the referee for this kind suggestion. We have added the pumping condition of the far-field measurements in the legends of Fig. 4 and Fig.5 in the revised manuscript.

REVIEWERS' COMMENTS

Reviewer #1 (Remarks to the Author):

The authors have replied to all the questions that the other reviewer and I raised in the previous review round. One of the biggest concerns, the unfair comparison of the topological and conventional lasers, is not seen in the manuscript anymore and has been converted into a comparison of the topological lasers with different parameters. Another argument, the relationship between the proposed design and high output power, becomes much clearer and convincing. In this sense, it is reasonable to keep 'high power' in the title. Therefore, now I recommend the publication of the manuscript in Nature Communications in the current form.

Reviewer #2 (Remarks to the Author):

The authors have answered my concerns. I don't have further questions regarding this paper.

Reply to Referee #1

Original general comment:

The authors have replied to all the questions that the other reviewer and I raised in the previous review round. One of the biggest concerns, the unfair comparison of the topological and conventional lasers, is not seen in the manuscript anymore and has been converted into a comparison of the topological lasers with different parameters. Another argument, the relationship between the proposed design and high output power, becomes much clearer and convincing. In this sense, it is reasonable to keep 'high power' in the title. Therefore, now I recommend the publication of the manuscript in Nature Communications in the current form.

Our reply:

We sincerely thank the referee for the positive comments on this work. And we are very much grateful for the insightful comments and suggestions that help us to improve the quality of this work.

Reply to Referee #2

Original general comment:

The authors have answered my concerns. I don't have further questions regarding this paper.

Our reply:

We thank the referee for the supportive and encouraging comments. We sincerely appreciate that the two reviewers can recognize our work and agree to publish it in Nature Communications.